# Tyrosine-targeted covalent inhibition of a tRNA synthetase aided by zinc ion

Hang Qiao[1], Mingyu Xia[1], Yiyuan Cheng[1], Jintong Zhou[1,2], Li Zheng[1], Wei Li ⬤ [3], Jing Wang ⬤ [1,2✉] &
Pengfei Fang ⬤ [1,2✉]

Aminoacyl-tRNA synthetases (AARSs), a family of essential protein synthesis enzymes, are attractive targets for drug development. Although several different types of AARS inhibitors have been identified, AARS covalent inhibitors have not been reported. Here we present five unusual crystal structures showing that threonyl-tRNA synthetase (ThrRS) is covalently inhibited by a natural product, obafluorin (OB). The residue forming a covalent bond with OB is a tyrosine in ThrRS active center, which is not commonly modified by covalent inhibitors. The two hydroxyl groups on the *o*-diphenol moiety of OB form two coordination bonds with the conserved zinc ion in the active center of ThrRS. Therefore, the *β*-lactone structure of OB can undergo ester exchange reaction with the phenolic group of the adjacent tyrosine to form a covalent bond between the compound and the enzyme, and allow its nitrobenzene structure to occupy the binding site of tRNA. In addition, when this tyrosine was replaced by a lysine or even a weakly nucleophilic arginine, similar bonds could also be formed. Our report of the mechanism of a class of AARS covalent inhibitor targeting multiple amino acid residues could facilitate approaches to drug discovery for cancer and infectious diseases.

[1] State Key Laboratory of Bioorganic and Natural Products Chemistry, Center for Excellence in Molecular Synthesis, Shanghai Institute of Organic Chemistry, University of Chinese Academy of Sciences, Chinese Academy of Sciences, Shanghai 200032, China. [2] School of Chemistry and Materials Science, Hangzhou Institute for Advanced Study, University of Chinese Academy of Sciences, Hangzhou 310024, China. [3] Department of Medicinal Chemistry, School of Pharmacy, China Pharmaceutical University, Nanjing 211198, China. ✉email: jwang@sioc.ac.cn; fangpengfei@sioc.ac.cn

Aminoacyl-tRNA synthetases (AARSs) are housekeeping enzymes catalyzing the formation of an ester bond between a specific amino acid and its cognate tRNA[1,2]. They undertake the essential function of protein synthesis in all cells, including pathogenic microorganisms. Despite their similarity across organisms, the sequence and topological differences between pathogenic microbial AARSs and human AARSs make it possible to design drugs that selectively inhibit pathogen AARSs[3]. AARSs suppression can be exploited for cancer chemotherapy since over-proliferating cancer cells are more sensitive to AARSs suppression than normal cells[4]. For these reasons, AARS are underexploited therapeutic targets[5].

Disproportionate to the vast potential targets offered by this family, few AARS inhibitors have yet been developed into drugs, including mupirocin, approved for the treatment of skin infections[6], AN2690, approved for the treatment of onychomycosis[7], and halofuginone, a veterinary drug used to treat coccidiosis, received orphan drug designation for treating systemic sclerosis[8]. More AARS inhibitors are being actively developed[9–11]. Different types of AARS inhibitors have been found, including substrate mimetics, Trojan horses, induced-fit inhibitors, and reaction hijacking inhibitors[12–17]. However, covalent inhibitors of AARS have not been reported. The compound closest to a covalent inhibitor may be the boron-containing compound AN2690, which covalently binds to the 2' and 3' hydroxyl groups of the ribose of the terminal adenylate (A76) of tRNA[Leu] and traps tRNA[Leu] at the editing site of leucyl-tRNA synthetase (LeuRS)[18]. There is no report of structural evidence of any inhibitors that directly form covalent bonds with an AARS enzyme itself.

Threonyl-tRNA synthetase (ThrRS) inhibitors such as borrelidin have been shown to have a wide range of biological activities, including antimicrobial, antimalarial, and antiangiogenic activities[19–21]. Recently, a novel natural product, obafluorin (OB), produced by *Pseudomonas fluorescens* ATCC 390502[22], was identified as a potent inhibitor of bacterial ThrRS[23]. OB is synthesized through the nonribosomal peptide synthetase (NRPS) assembly line[24–27], and contains a $\beta$-lactone ring, showing a novel structure compared to all antibiotics approved by the US Food and Drug Administration (FDA). The mechanism of action of OB is unknown.

Here we report the mechanism of action of OB, a covalent inhibitor of an AARS, and show that $\beta$-lactone can covalently modify tyrosine, lysine, and arginine residues on proteins, which will be helpful for the design and development of covalent inhibitors targeting AARSs.

## Results

**Crystal structure of the OB–ThrRS complex.** OB contains a $\beta$-lactone ring with a 2,3-dihydroxybenzamidyl moiety attached to the $\alpha$ position and a 4-nitrobenzyl moiety attached to the $\beta$ position (Fig. 1a). It shows no apparent structural similarities with the three substrates of ThrRS: L-threonine, ATP, and tRNA. Accordingly, molecular docking failed to predict a plausible binding mode between OB and ThrRS (Supplementary Fig. 1). To elucidate the mechanism by which OB inhibits ThrRS, we cocrystallized a fragment of *Escherichia coli* ThrRS containing catalytic and anticodon-binding domains (residues 242–642, Fig. 1b) with OB and determined the structure to a resolution of 2.5 Å (Supplementary Table 1). One asymmetric unit contained a ThrRS dimer (Fig. 1c), which is the typical oligomeric state of class II AARSs[28]. In the two catalytic domains of ThrRS homodimer, OB was bound to the active site for Thr-AMP formation (Fig. 1c, d). In total, 18 residues in the active center of ThrRS were located within 4.5 Å of OB and thus interacted with it (Fig. 1e).

**OB forms a covalent bond with Tyr462.** Interestingly, the four-membered ring of the $\beta$-lactone of OB was opened in the ThrRS-OB structure. Instead, the acyl group formed a new ester bond with the phenolic group of Tyr462 (Fig. 2a and Supplementary Fig. 2). In addition, the two hydroxyl groups on the *o*-diphenol moiety of OB formed two coordination bonds with the conserved catalytic $Zn^{2+}$ ion of ThrRS (Fig. 2b). This structure suggests that OB inhibits ThrRS by forming coordination bonds with the $Zn^{2+}$ ion in the catalytic center of ThrRS, allowing its $\beta$-lactone ring to be approached and attacked by the phenolic group of Tyr462, and finally forming a new covalent bond with Tyr462 (Fig. 2c).

**OB blocks the binding of L-threonine and tRNA.** When ThrRS catalyzes tRNA aminoacylation, the $Zn^{2+}$ ion recognizes and binds L-threonine, assisting in the L-threonine activation reaction[29]. In addition, Asp383 forms an H-bond with the $\beta$-hydroxyl group of L-threonine; Gln381 and Gln484 form H-bonds with the carboxyl group of L-threonine; and Tyr462 forms an H-bond with the amino group (Supplementary Fig. 3a). The ThrRS-OB structure showed that the *o*-diphenol group of OB occupied the position of L-threonine and replaced it to bind $Zn^{2+}$ ion and Asp383. At the same time, the conformations of Gln484 and Met332 were changed to accommodate OB (Fig. 3a).

On the other hand, the opening of the $\beta$-lactone ring prolonged OB's molecular configuration. This allowed its nitrobenzene group to extend to the other side of the pocket and form stacking interactions with Tyr313 and Arg363, two important residues for ThrRS to bind the 3' end of tRNA[30] (Supplementary Fig. 3b). Therefore, the nitrobenzene group of OB occupied the binding site of tRNA A76, which prevented tRNA from entering the active center of ThrRS (Fig. 3b).

**OB covalently modifies engineered lysine.** $\beta$-Lactones are ring-strained compounds that function as effective acylating agents when nucleophiles are present. We hypothesized that if Tyr462 was mutated to a lysine residue, the mutant protein ThrRS_Y462K should also be covalently inhibited by OB, because these two residues have similar side chain lengths and lysine has stronger nucleophilicity.

To test this hypothesis, we purified the ThrRS_Y462K mutant, and developed a thermal shift assay (TSA) which can rapidly evaluate the effect of the inhibitor. We used compound 36j as a positive control (Supplementary Fig. 4) because it binds to all three substrate sites of *Salmonella enterica* ThrRS with a dissociation constant ($K_D$) of ~35 nM, while also showing strong inhibition of *E. coli* ThrRS[31]. In the TSA experiment, 36j increased the mid-melting point (Tm) of ThrRS_WT by 32.1 °C, and increased the Tm of ThrRS_Y462K by 34.6 °C (Fig. 4a and Supplementary Fig. 5a, b). OB increased the Tm of ThrRS_WT by 34.8 °C (Fig. 4a and Supplementary Fig. 5a), which was 2.7 °C more than that of 36j. Interestingly, OB also increased the Tm of ThrRS_Y462K by 36.8 °C (Fig. 4a and Supplementary Fig. 5b), suggesting that OB forms a covalent bond with the engineered Lys462.

To visualize the covalent bond, we determined the 1.9 Å structure of ThrRS_Y462K in the presence of OB (Supplementary Table 2). The overall structure of the ThrRS_Y462K mutant was very similar to that of the wild-type protein (Fig. 4b). As predicted, Lys462 also induced the ring-opening and covalent bond formation of the $\beta$-lactone. At the same time, coordination bonds were formed between the *o*-diphenol group and the $Zn^{2+}$ ion, and stacking interactions were maintained between the nitrobenzene group and ThrRS Tyr313 and Arg363 residues (Fig. 4c, d and Supplementary Fig. 6).

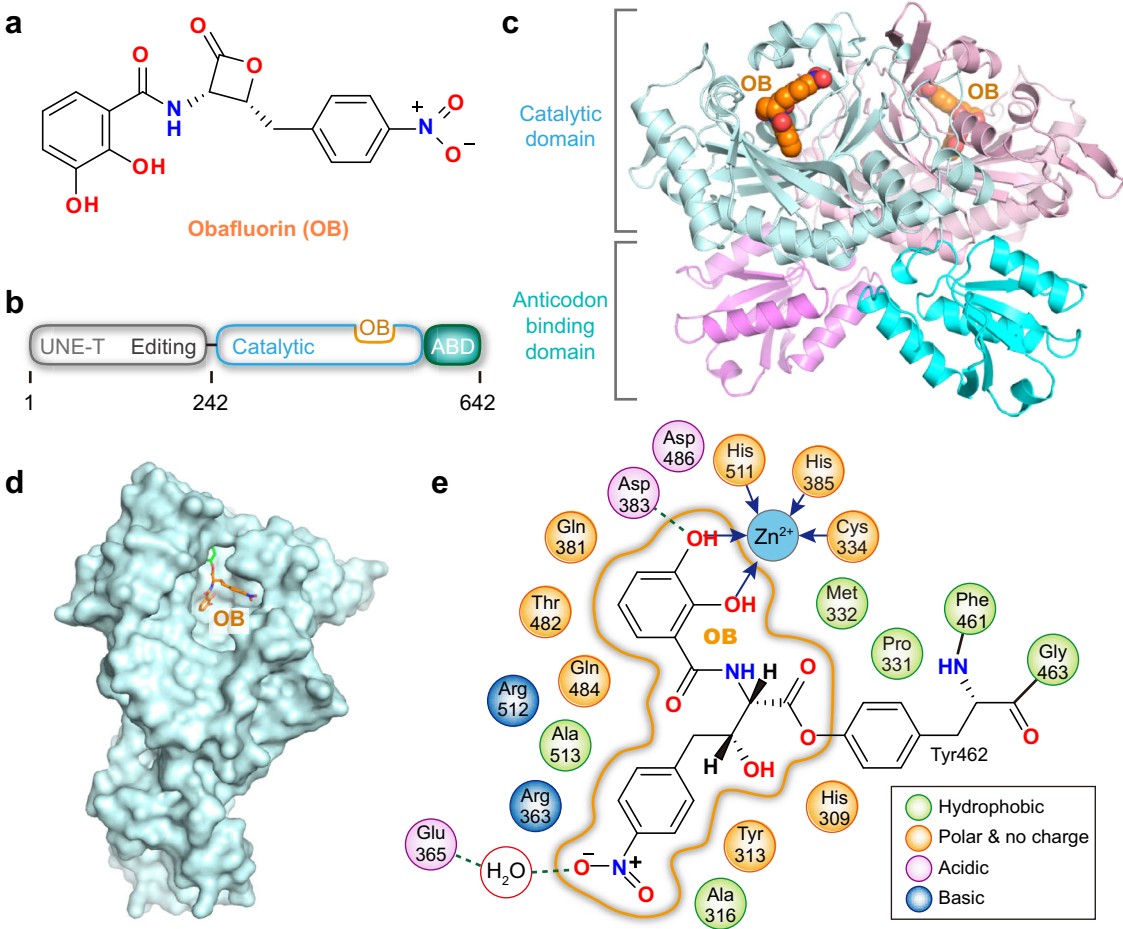

**Fig. 1 Structure of *E. coli* ThrRS bound to Obafluorin (OB). a** Chemical structure of OB. **b** Schematic representation of the domain organization of *E. coli* ThrRS. ABD anticodon-binding domain. **c** Two ThrRS monomers form an asymmetric unit that is a homodimer. OB is shown as spherical models at the active site of both subunits. **d** OB (orange sticks) is bound at the center of the ThrRS active pocket (light cyan surface). **e** Two-dimensional presentation of the OB binding site. Residue Tyr462 is covalently linked to OB through an ester bond. Coordination bonds are shown as blue arrows. H-bonds are shown as green dashed lines.

**OB prevents ATP binding to ThrRS**. In both ThrRS_WT-OB and ThrRS_Y462K-OB complex structures, OB occupied only the binding sites for L-threonine and tRNA A76, leaving the ATP-binding site vacant (Supplementary Fig. 7a, b). To check whether ThrRS could bind OB and ATP simultaneously, we cocrystallized ThrRS_Y462K with OB in the presence of ATP, and solved the structure to a resolution of 2.2 Å (Supplementary Table 3). The ATP-binding pocket remained ATP-free in the new ThrRS_Y462K-OB structure (Supplementary Fig. 7c), suggesting that the OB–ThrRS interaction prevents ATP from entering its pocket. When superimposing the ThrRS_Y462K-OB structure onto the previously solved ThrRS-ATP complex structure, no strong clash between OB and ATP is exhibited. However, the distances between the alpha phosphate of ATP and the two benzene rings of OB are 2.8 and 4.2 Å (Supplementary Fig. 7d). Therefore, there may be repulsion between the hydrophilic groups of ATP and the hydrophobic groups of OB.

To test the mutually exclusive effect between OB and ATP, we engineered a ThrRS_Y462F mutant that mutated Tyr462 to phenylalanine to avoid covalent bonding with OB. As a result, in the TSA experiment, OB only increased the Tm of ThrRS_Y462F by 2.6 °C, while 36j still increased the Tm of ThrRS_Y462F by 29.7 °C (Fig. 5a and Supplementary Fig. 5c). Consistently, OB failed to inhibit the activity of ThrRS_Y462F in the ATP hydrolysis assay up to 5 µM (Fig. 5b). More interestingly, when

we cocrystallized ThrRS_Y462F with OB and ATP (Supplementary Table 4), ATP was visible in the active site of ThrRS together with two cobound Mg²⁺ ions (Fig. 5c). No densities for OB were observed in the active center (Supplementary Fig. 8a). Without the binding of OB, the conformation of Tyr313 of ThrRS_Y462F was in a position that would clash with OB as it was in the ThrRS-OB complexes (Supplementary Fig. 8b). In addition, compared with the ThrRS_Y462F-ATP structure, OB induced a significant conformational change to the outer side of the active cleft (Supplementary Fig. 8c), which was similar to that induced by another potent ThrRS inhibitor, borrelidin[16] (Supplementary Fig. 8d). Therefore, OB prevents ThrRS from binding to all three substrates, including L-threonine, tRNA, and ATP.

**OB modifies arginine after prolonged incubation**. Because the length of the side chain of arginine is also not that different from tyrosine, we designed a ThrRS_Y462R mutant. Although the guanidinium group of arginine has less inclination for the donation of electron density due to resonance, we wondered if the β-lactone could covalently modify arginine when it was properly positioned.

Not surprisingly, OB only increased the Tm of ThrRS_Y462R by 6.5 °C in the TSA experiment, which was higher than that of ThrRS_Y462F but significantly lower than that of ThrRS_WT or ThrRS_Y462K (Fig. 5a and Supplementary Fig. 5d). This result

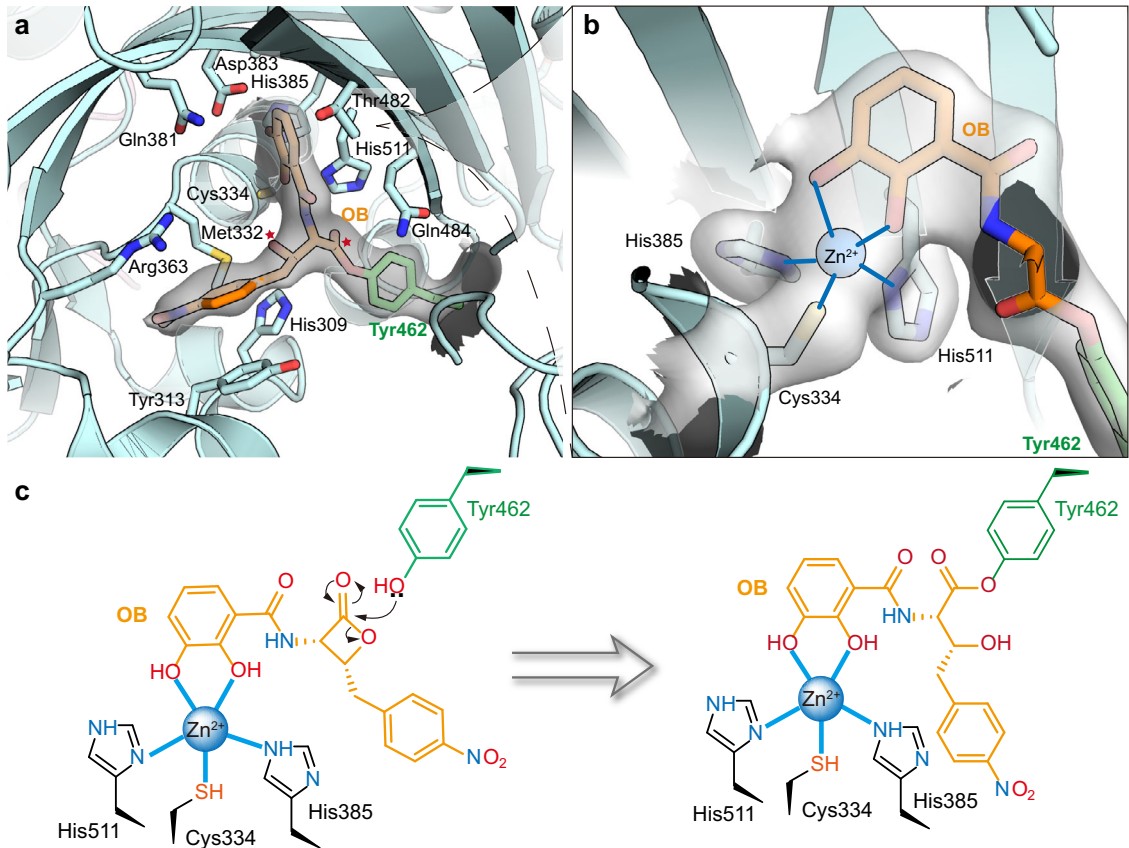

**Fig. 2 OB forms a covalent bond with the Tyr462 residue of ThrRS. a** The catalytic center of ThrRS with bound OB. The phenolic group of Tyr462 (green sticks) forms a new ester bond with OB (orange sticks). The linkage atoms of the original 4-membered ring in OB are indicated by red stars. Interacting residues are shown as sticks. The 2Fo-Fc electron density of Tyr462-OB (contoured at 1.0 σ) is shown as a gray transparent surface. **b** Close-up view of residues coordinating with a $Zn^{2+}$ ion. Two hydroxyl groups on the *o*-diphenol moiety of OB coordinate with a $Zn^{2+}$ ion. Residues Cys334, His385, His511, Tyr462, and OB are shown as sticks. The 2Fo-Fc electron density of these residues (contoured at 1.0 σ) is shown as a transparent surface. **c** Schematic illustration of the covalent bond formation between ThrRS (Tyr462) and OB.

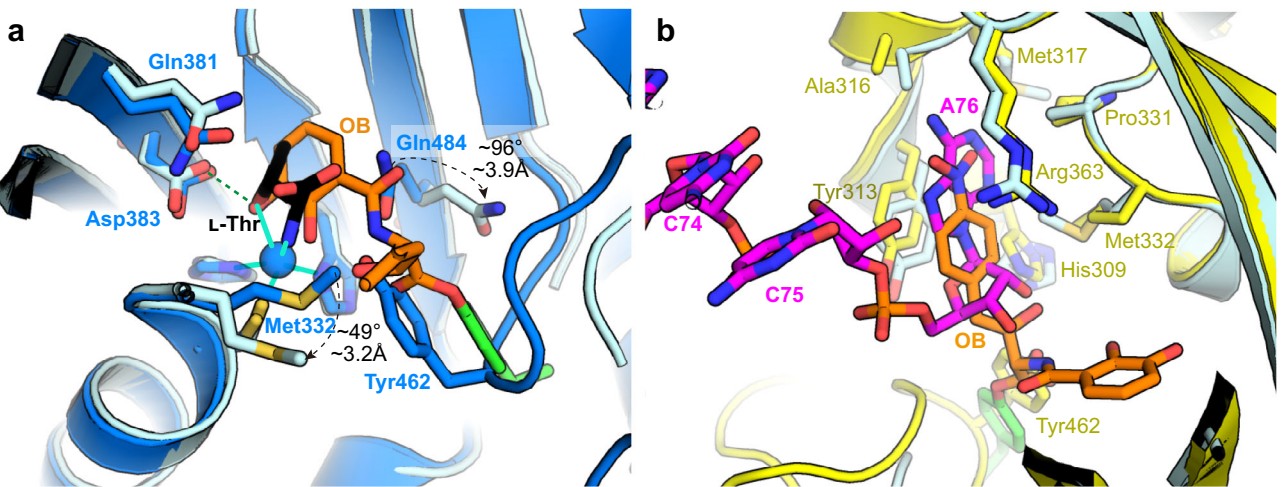

**Fig. 3 OB precludes ThrRS binding of ʟ-threonine and tRNA. a** Superimposition of the ThrRS-ʟ-Thr structure (blue cartoons, PDB code: 1EVK) with the ThrRS-OB structure (light cyan cartoons). OB is shown as orange sticks. ʟ-Thr is shown as black sticks. H-bonds are shown as green dashed lines. Coordination bonds are shown as cyan lines. **b** Superimposition of the ThrRS-tRNA^Thr structure (yellow cartoons, PDB code: 1QF6) with the ThrRS-OB structure (light cyan cartoons). The nitrophenyl group forms stacking interactions with Tyr313 and Arg363, which also interact with tRNA A76 in a similar way.

suggests that Arg462 could not react with β-lactone during the relatively short period of time in the TSA experiment.

When we cocrystallized ThrRS_Y462R with OB and ATP (Supplementary Table 5), we expected that ATP would exclude OB from the active pocket of ThrRS_Y462R if OB failed to form a covalent bond with the enzyme, as we have observed in the ThrRS_Y462F structure (Fig. 5c). However, the result was surprising in that Arg462 actually formed a covalent bond with

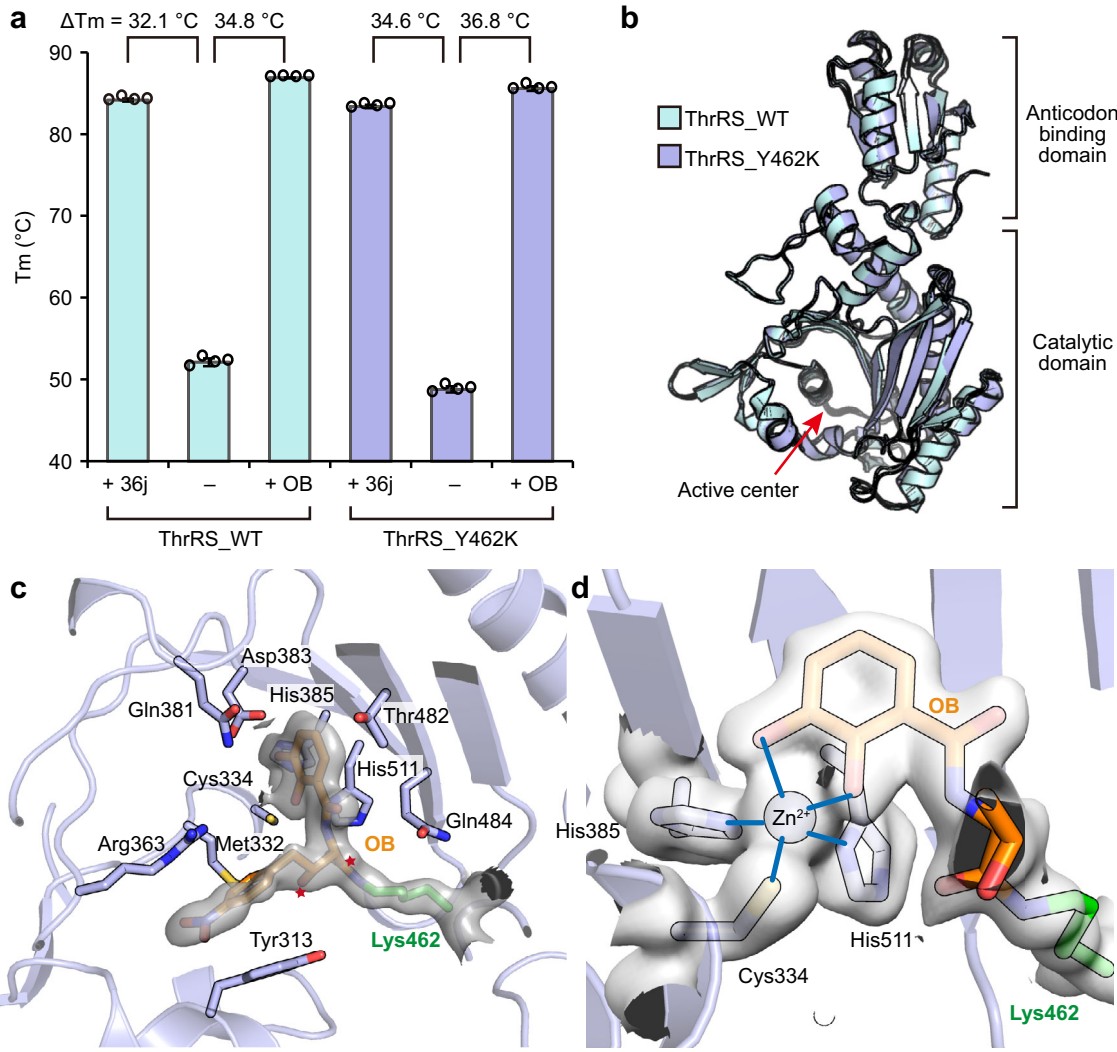

**Fig. 4 OB covalently modifies the engineered Lys462. a** Diagram of the Tm value of ThrRS_WT and ThrRS_Y462K in the presence or absence of OB and 36j. Evaluations were carried out in four repeats, and error bars indicate the respective standard deviation ($n = 4$, mean value ± SD). All data points are shown in small circles. Numerical Data can be found in Supplementary Data 1. **b** Superimposition of the structures of ThrRS_WT-OB and ThrRS_Y462K-OB. The r.m.s.d is 0.421 Å over 322 Cα atoms. **c** The catalytic center of ThrRS_Y462K bound to OB. The ε-amino group of Lys462 (presented as green sticks) forms an amide bond with OB (presented as orange sticks). The linkage atoms of the original 4-membered ring in OB are indicated by red stars. Interacting residues are shown as sticks. The 2Fo-Fc electron density of Lys462-OB (contoured at 1.0 σ) is shown as a transparent surface. **d** Close-up view of residues coordinating with a $Zn^{2+}$ ion. The 2Fo-Fc electron density of the involved residues (contoured at 1.0 σ) is shown as a transparent surface.

OB (Fig. 5d and Supplementary Fig. 9), while no ATP was observed in the pocket (Supplementary Fig. 10). This result indicates that although the activity of the arginine mutant is reduced, it can still react with β-lactone to form a covalent bond over a longer period of time, as it takes approximately 24 h for crystal growth.

## Discussion
Advances in covalent drug discovery have led to successful drugs, including inhibitors of epidermal growth factor receptor (EGFR), Bruton's tyrosine kinase (BTK), KRAS(G12C), and SARS-CoV-2 main protease[32]. An appropriate warhead is a key point in the development of covalent inhibitors.

This work shows that OB uses its β-lactone moiety to covalently modify Tyr462 at the active site of ThrRS. It has been found that cysteine, serine, and threonine can be covalently modified by β-lactones. For example, hymeglusin inhibits eukaryotic hydroxymethylglutaryl-CoA synthase (HMGCS) by forming a thioester adduct to the active site cysteine[33]. Orlistat covalently

binds to serine residues at the active site of lipase and is approved by the FDA for the treatment of obesity[34]. Salinosporamide A (marizomib), a hybrid polyketide-nonribosomal peptide (PK-NRP), is a potent 20 S proteasome inhibitor, and is approved by the FDA as an orphan drug for the treatment of multiple myeloma. Its β-lactone ring covalently modifies the N-terminal threonine of the 20 S proteasome[35]. The phenolic group of tyrosine is less nucleophilic than the hydrosulfonyl or regular hydroxyl groups that have been reported to attack lactones. On the other hand, fluorosulfonyl groups are currently one of the few, if not the only, warheads that covalently modify tyrosine[36]. The discovery that OB can form a covalent bond with Tyr462 of ThrRS provides a design strategy for the development of covalent inhibitors targeting tyrosine.

Another discovery in this study is that β-lactone can modify arginine, although it is not highly reactive. Arginine is a common residue in the active pocket of enzymes that use nucleotide molecules as substrates, such as AARSs and glycosyltransferases. Currently, there are few strategies to develop covalent inhibitors

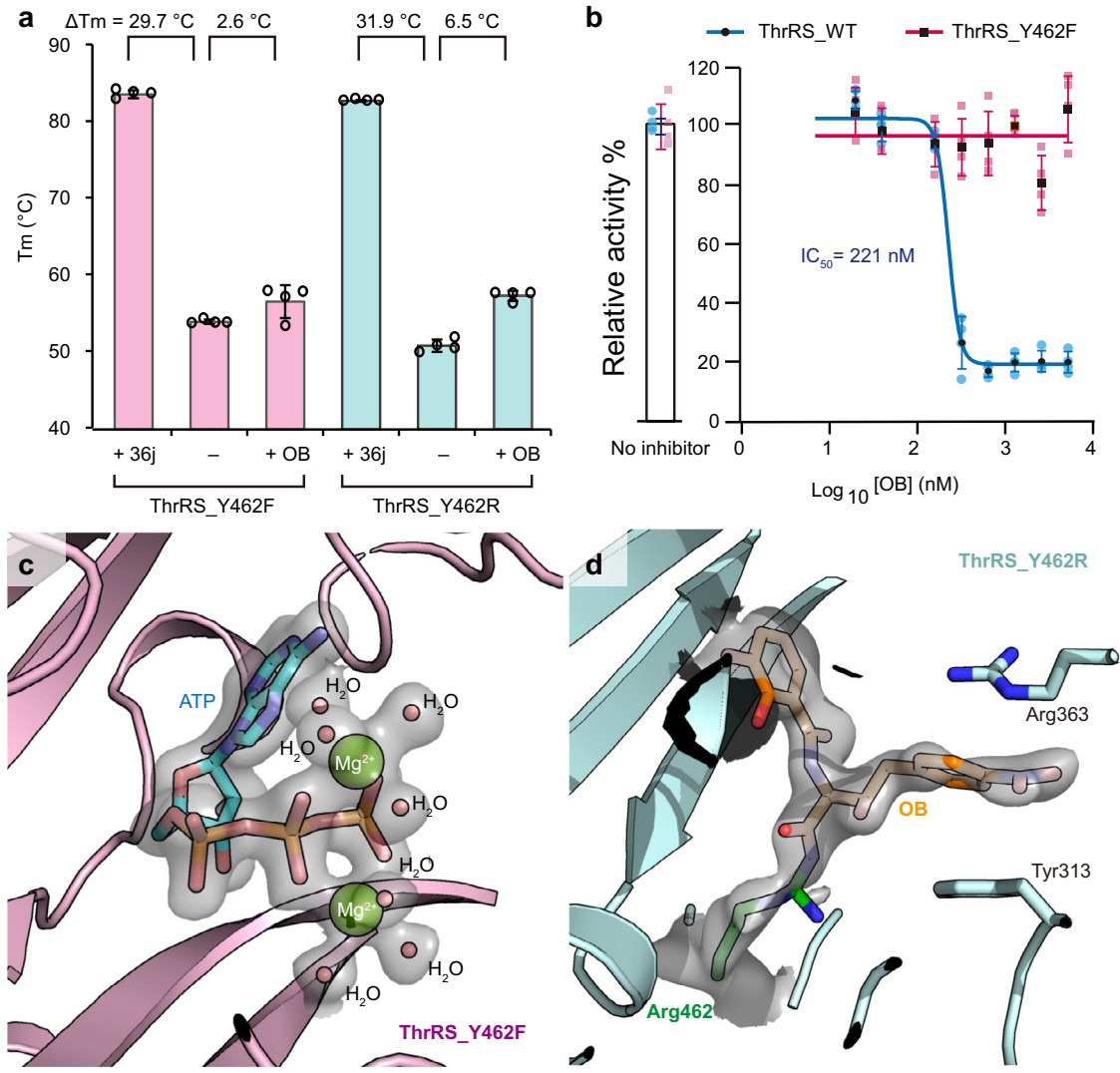

**Fig. 5 OB binds to ThrRS_Y462R but not the Y462F mutant. a** Diagram of the Tm value of ThrRS_Y462F and ThrRS_Y462R in the presence or absence of OB and 36j. Evaluations were carried out in four repeats, and error bars indicate the respective standard deviation ($n = 4$, mean value ± SD). All data points are shown in small circles. Numerical Data can be found in Supplementary Data 2. **b** Inhibitory curves of OB on the ATP hydrolysis activity of ThrRS_WT or ThrRS_Y462F. Evaluations were carried out in four repeats, and error bars indicate the respective standard deviation ($n = 4$, mean value ± SD). All data points for ThrRS_WT and ThrRS_Y462F are shown in pale blue dots and pale pink square dots, respectively. Numerical Data can be found in Supplementary Data 3. **c** Zoomed-in view of the ATP-binding site of ThrRS_Y462F. The 2Fo-Fc electron density of the bound ATP and cobound $Mg^{2+}$ ions and water molecules is contoured at 1.0 σ and shown as a transparent surface. ATP is shown as sticks. $Mg^{2+}$ ions and water molecules are shown as spheres. **d** Zoomed-in view of the OB binding site of ThrRS_Y462R. The 2Fo-Fc electron density of Arg462-OB (contoured at 0.8 σ) is shown as a transparent surface.

targeting arginine[37]. Covalent modifications with lower activity tend to have higher selectivity. Therefore, β-lactone's slow modification of arginine may be also useful for the development of covalent inhibitors.

In summary, this work not only reports OB as an AARS covalent inhibitor, but also shows that if placed in a suitable position, β-lactone could covalently modify tyrosine, lysine, and even the weakly nucleophilic residue arginine, which are not commonly used to develop covalent inhibitors, in addition to previously reported cysteine, serine, and threonine.

## Methods

**Molecular docking**. Molecular docking was done by AutoDock Vina[38]. *E. coli* ThrRS (receptor, PDB code: 1EVK) and Obafluorin (OB, ligand) were prepared using AutoDockTools (v1.5.7). For the preparation of receptor input files, all water molecules and ligands were removed, and polar hydrogens were added. The nine

highest-scoring poses of OB were output. All visualizations were done using PyMOL (https://pymol.org/).

**Protein preparation**. The C-terminal 6×His-tagged *E. coli* ThrRS (residues 242–642) and Y462F/K/R mutants were constructed in a pET28a vector respectively. Each protein was expressed in BL21 (DE3) strain and induced with 0.5 mM isopropyl-β-D-thiogalactoside for 20 h at 16 ℃. The cell pellet (from 2 to 4 liters) was lysed in buffer A (25 mM Tris pH 7.5, 500 mM NaCl, and 25 mM imidazole), loaded onto a Ni-Hitrap column (Cytiva, USA), and washed with buffer A, then eluted with buffer B (25 mM Tris pH 7.5, 500 mM NaCl, and 250 mM imidazole). The eluted protein was further purified by a Hitrap Q HP anion exchange column (Cytiva, USA) with NaCl gradient (0.05–1 M NaCl in 25 mM Tris pH 7.5). The peak fraction was further purified by gel filtration S200 (Cytiva, USA) with a buffer containing 25 mM Tris pH 7.5, 200 mM NaCl, and 1 mM MgCl₂. The final purified protein was used for crystallization immediately. The rest protein was flash-frozen by liquid nitrogen and stored at −80 ℃.

**Crystallization and structure determination**. All crystallization experiments were performed at 18 ℃ based on the sitting-drop method. All proteins were

concentrated to 13–15 mg/mL using a 10 kDa centrifugal filter (Millipore, USA). Before crystallization, OB was mixed with protein at twice the molar ratio, and ATP was added at 5 times the molar ratio when it was used. For microbatch crystallization screen, 0.5 µL protein solution was mixed with an equal amount of precipitant solution (Molecular dimensions, UK) in microbatch 96-well plates using a Gryphon robot (ART technology, USA). Crystals grew to final dimensions within 1–3 days.

The ThrRS_WT–OB crystals were obtained from the condition of 0.03 M diethylene glycol, 0.03 M triethylene glycol, 0.03 M tetraethylene glycol, 0.03 M pentaethylene glycol, 0.05 M sodium HEPES, 0.05 M MOPS acid pH 7.5, 20% v/v ethylene glycol, and 10% w/v PEG 8000.

The ThrRS_Y462K–OB (no ATP) crystals were obtained from the condition of 2.0 M Ammonium sulfate, and 0.15 M sodium citrate pH 5.5.

The ThrRS_Y462K–OB (with ATP) crystals were obtained from 0.03 M diethylene glycol, 0.03 M triethylene glycol, 0.03 M tetraethylene glycol, 0.03 M pentaethylene glycol, 0.05 M sodium HEPES, 0.05 M MOPS acid pH 7.5, 20% v/v ethylene glycol, 10% w/v PEG 8000.

The ThrRS_Y462F–ATP crystals were obtained from 0.03 M diethylene glycol, 0.03 M triethylene glycol, 0.03 M tetraethylene glycol, 0.03 M pentaethylene glycol, 0.045 M imidazole, 0.055 M MES monohydrate acid pH 6.5, 20% v/v ethylene glycol, and 10% w/v PEG 8000.

The ThrRS_Y462R–OB crystals were obtained from 0.02 M DL-glutamic acid monohydrate, 0.02 M DL-alanine, 0.02 M glycine, 0.02 M DL-lysine monohydrochloride, 0.02 M DL-serine, 0.05 M sodium HEPES, 0.05 M MOPS acid pH 7.5, 20% v/v ethylene glycol, and 10% w/v PEG 8000.

The resulting crystals were flash-frozen in liquid nitrogen for data collection. Data sets were obtained from beamline 10U2 at Shanghai Synchrotron Radiation Facility (SSRF). Then data sets were indexed and integrated with XDS[39] or auto-processed by AutoPROC[40] & Xia2_dials[41] at Shanghai Synchrotron Radiation Facility (SSRF). The HKL files were scaled and merged with Aimless in CCP4 suite[42]. The structures were determined by molecular replacement using E. coli ThrRS structure (PDB code: 1FYF) as a search model in the program Molrep in CCP4 suite[43]. After corrections for bulk solvent and overall B values, data were refined by iterative cycles of positional refinement and TLS refinement with PHENIX[44] and model building with COOT[45]. All current models have good geometry and no residues are in the disallowed region of the Ramachandran plot. Data collection and model statistics are given in Supplementary Table 1–5.

**Thermal shift assay**. ThrRS_WT and Y462F/K/R mutant proteins were prepared at 2 µM concentration in a buffer containing 25 mM Tris-HCl pH 7.5, 200 mM NaCl, and 20 µM OB (GlpBio, UK) or ddH$_2$O in an equal volume since compounds were diluted by ddH$_2$O to 1 mM as stocks. Compound 36j was assayed at the same final concentration as a positive control. SYPRO Orange dye (Sigma, USA) was diluted in the assay buffer containing 25 mM Tris-HCl pH 7.5 and 200 mM NaCl to a 40× concentration, and was added in the mixture to a final 4× concentration. Aliquots (20 µL) were added to a 96-well PCR plate. After complete mixing, the final solutions were heated from 25 to 95 °C at a rate of 0.015 °C/s, and fluorescence signals were monitored by QuantStudio 3 (Applied Biosystems by Thermo Fisher Scientific, USA).

**ATP hydrolysis assay**. ATP hydrolysis assay was based on Kinase-Glo® luminescent Kit (Promega, USA). 200 nM ThrRS_WT or ThrRS_Y462F was incubated with serial diluted OB (0 to 10 µM) in a buffer containing 25 mM HEPES pH 7.5, 50 mM NaCl, 40 mM MgCl$_2$, 30 mM KCl, 0.01 mg/mL bovine serum albumin (BSA), and 0.004% Tween-20 at 37 °C for 4 h, then 1:1 mixed with the substrates mixture containing 4 µM ATP, 40 µM L-threonine, 25 mM HEPES pH 7.5, 50 mM NaCl, 40 mM MgCl$_2$, 30 mM KCl, 0.01 mg/mL BSA, and 0.004% Tween-20. ATP hydrolysis reaction was performed at 37 °C for 4 h. Then the detection solution was added into the reaction system at a 1:1 ratio and gently shaken for 10 min. The chemiluminescence signal was measured using a microplate reader (Tecan, USA). All experiments were performed in four replicates. Data were processed using GraphPad Prism 8.

**Statistics and reproducibility**. Enzymatic assay and thermal shift measurements were conducted in four repeats. Acquired data are presented as the mean values ± standard deviation (SD).

**Reporting summary**. Further information on research design is available in the Nature Portfolio Reporting Summary linked to this article.

## Data availability

Data supporting the findings of this study are available within the article and its supplementary information files. Atomic coordinates and structure factors for the reported crystal structures have been deposited with the Protein Data Bank under accession numbers 8H98, 8H99, 8H9A, 8H9B, and 8H9C.

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

## Acknowledgements

We thank Prof. Biao Yu and Zhen Wang for their helpful discussion and Prof. Huihao Zhou for his supply of compound 36j. We gratefully acknowledge the help from the staff of beamline 10U2 at Shanghai Synchrotron Radiation Facility. This work is supported by the National Key Research and Development Program of China grant 2022YFC2303100, National Natural Science Foundation of China grants 21977107, 22277132, 21977108, 22277134, and 21977115, Shanghai Science and Technology Committee grant 22ZR1475000, and the State Key Laboratory of Bioorganic and Natural Products Chemistry.

## Author contributions

Conceptualization: H.Q. and P.F. Methodology: H.Q., M.X., Y.C., J.Z., W.L., J.W., and P.F. Investigation: H.Q., M.X., Y.C., and J.Z. Visualization: H.Q., J.W., and P.F. Funding acquisition: W.L., J.W., and P.F. Project administration: L.Z., W.L., J.W., and P.F. Supervision: W.L., J.W., and P.F. Writing—original draft: H.Q. and P.F. Writing—review and editing: W.L., J.W., and P.F.

## Competing interests

The authors declare no competing interests.
