## [Peer Review File · Communications Biology]

Reviewers' comments:

Reviewer #1 (Remarks to the Author):

Aminoacyl tRNA synthetases (AARSs) are well proven antibacterial targets. Many different types of AARSs Inhibitors including substrate mimetics, Trojan horses, induced-fit, and reaction hijacking have been reported. However, no inhibitors have been found to form direct covalent interactions with the residues in the active site cavity of AARSs. In this manuscript, Qiao et al identified obafuorin (OB), a natural product inhibitor newly identified to inhibit bacterial ThrRS, as the first covalent inhibitor for AARS family members. The cocrystal structure of OB with ThrRS revealed that OB forms an unusual covalent connection with tyrosine through a β -lactone structure transesterification reaction. The TSA binding assay and cocrystal structures of tyrosine-mutated ThrRSs further highlighted the β -lactone as a powerful covalent warhead which can target multiple different amino acids. In all, the new inhibitory mechanism disclosed in this study is of exceptional novelty and importance; the technical work is impeccable; the conclusion based on high quality X-ray structures and mutation analysis is convincing; the data presenting is concise and clear. Therefore, this reviewer strongly recommends the rapid publication of this manuscript in Communications Biology without delay.

The only suggestions this reviewer can make to the authors are quite minor:

1.P 3, line 13. "halofuginone, approved for scleroderma as an orphan drug [7]". As I know, halofuginone has received orphan drug designation for the treatment of systemic sclerosis, but has not yet been approved by FDA to enter the market. So far, halofuginone can be only used as a veterinary drug for treating coccidiosis.

In Figures 4a and 5a, it is suggested to present wt and mutant ThrRS proteins in the same colors as their cartoon structures.

Reviewer #2 (Remarks to the Author):

The major claim of the paper is that a new class AARS inhibitor has been identified. The authors' insistence on referring to a powerful warhead of covalent inhibitors throughout the manuscript detracts from otherwise solid science. This work would be of interest to the broader community, and the work requires revisions to be publishable.

Abstract

page 2 line 4 change discovered to identified

page 2 lines 13 - 16 reads better as Our report of the mechanism of a new class of AARS covalent inhibitor targeting multiple amino acid residues could facilitate new approaches to drug discovery for cancer and infectious diseases.

Introduction

Please rewrite and remove all the unnecessary extra "the"

For accuracy, correct lines 7 - 10 on page 3.

AARS suppression can be exploited for cancer chemotherapy since over-proliferating cancer cells are more sensitive to AARS suppression than normal cells [4]. For these reasons, AARS are underexploited therapeutic targets [5].

page 4, line 5 -6

Fix the last sentence. You report more than the mechanism of OB.

Results section headers use correct tense:

page 4 line 9

Crystal structure of the OB - ThrRS complex

page 4 line 23

OB forms a covalent bond with Tyr462

page 5 line 9

OB blocks the binding of ...

page 5 line 24
OB covalently modifies engineered lysine

page 6 line 23
OB prevents ATP binding to ThrRS

page 8 line 4
OB modifies arginine after prolonged incubation. What is the significance? Rewrite the heading to match.

Change your figures with electron density because 1.0 and 0.8 $2F_o-F_c$ are insufficient. Instead, present omit maps at 3.0 σ and as mesh.

Update all figure legends to the present tense
for example, figure 4 reads better as "OB covalently modifies engineered Lys 462."

Other minor changes are required, like the improper use of uppercase in the names of compounds in the materials and methods section.

Do you want to use active voice or passive voice? Be consistent in your sentences and not jump from one to another.

I have attached a pdf file with additional handwritten comments.

RE: manuscript COMMSBIO-22-3988-T by Qiao *et al.*

Reviewers provided thoughtful and helpful comments on our work. We considered and discussed these comments very carefully. Listed below are point-by-point responses to each comment of each reviewer.

Reviewer #1 highly praised our work, recommended the publication of this manuscript, and put forward two revision suggestions.

1. P 3, line 13. “halofuginone, approved for scleroderma as an orphan drug [7]”. As I know, halofuginone has received orphan drug designation for the treatment of systemic sclerosis, but has not yet been approved by FDA to enter the market. So far, halofuginone can be only used as a veterinary drug for treating coccidiosis.

An: We thank the reviewer for pointing out this error, and we have revised the statement of halofuginone accordingly (*Page 3 line 12–13*).

2. In Figures 4a and 5a, it is suggested to present wt and mutant ThrRS proteins in the same colors as their cartoon structures.

An: We have adopted the reviewer's suggestion and used the same colors as the cartoon structures for the corresponding bar diagrams of wt and mutant proteins in *Fig. 4a* and *Fig. 5a*.

Reviewer #2 believed that this paper identified a new class AARS inhibitor of broad interest, suggested weakening the emphasis on the covalent warhead, and kindly offered detailed suggestions for revisions.

An: We appreciate the reviewer's suggestions, which we have fully adopted and made corresponding revisions in the paper. All changes are marked in blue in the text.

Abstract

page 2 line 4 change discovered to identified

page 2 lines 13 - 16 reads better as Our report of the mechanism of a new class of AARS covalent inhibitor targeting multiple amino acid residues could facilitate new approaches to drug discovery for cancer and infectious diseases.

An: We have made the suggested revisions *to Page 2 line 4 and line 13–16*.

Introduction

Please rewrite and remove all the unnecessary extra "the"

An: We have done our best to identify and remove unnecessary "the".

For accuracy, correct lines 7 - 10 on page 3.

AARS suppression can be exploited for cancer chemotherapy since over-proliferating cancer cells are more sensitive to AARS suppression than normal cells [4]. For these reasons, AARS are underexploited therapeutic targets [5].

An: We have made the suggested revision to *Page 3 line 7–10*.

page 4, line 5 -6

Fix the last sentence. You report more than the mechanism of OB.

An: We have added a description of the properties of β -lactone. This sentence now reads as: "*Here we report the mechanism of action of OB, the first covalent inhibitor of an AARS, and show for the first time that β -lactone can covalently modify tyrosine, lysine, and arginine residues on proteins, which will be helpful for the design and development of covalent inhibitors targeting AARSs.*"

Results section headers use correct tense:

page 4 line 9

Crystal structure of the OB - ThrRS complex

page 4 line 23

OB forms a covalent bond with Tyr462

page 5 line 9

OB blocks the binding of ...

page 5 line 24

OB covalently modifies engineered lysine

page 6 line 23

OB prevents ATP binding to ThrRS

An: We have made the suggested revisions to all these section subheadings.

page 8 line 4

OB modifies arginine after prolonged incubation. What is the significance? Rewrite the heading to match.

An: We have amended the subheading to "*OB modifies arginine after prolonged incubation*" as suggested by the reviewer.

Arginine is a common residue in the active pocket of enzymes that use nucleotide molecules as substrates, such as AARSs, glycosyltransferases. Currently, there are few strategies to develop covalent inhibitors targeting arginine. This study shows that β -

lactone is a potential new method to covalently target arginine. Covalent modifications with lower activity tend to have higher selectivity. Therefore, β -lactone's slow modification of arginine may be useful for the development of covalent inhibitors.

We have added this discussion to *Page 9 line 18–23*.

Change your figures with electron density because 1.0 and 0.8 $2F_o-F_c$ are insufficient. Instead, present omit maps at 3.0 σ and as mesh.

An: We have added the omit maps in *Supplementary Fig. S2, S6, and S9*.

Update all figure legends to the present tense

for example, figure 4 reads better as "OB covalently modifies engineered Lys 462."

An: We have updated all figure legends to the present tense in the revised manuscript.

Other minor changes are required, like the improper use of uppercase in the names of compounds in the materials and methods section.

Do you want to use active voice or passive voice? Be consistent in your sentences and not jump from one to another.

I have attached a pdf file with additional handwritten comments.

An: We apologize for the language problems in the original manuscript. We have corrected the improper uppercase in the names of compounds, and revised the manuscript to avoid sudden switching between active voice and passive voice.

We have also revised the manuscript according to the attached pdf file.

After these changes, we find that the manuscript reads much better. Many thanks to the reviewer for the generous guidance on the writing of the paper.